# Magnetic Janssen effect

L. Thorens [1,2], K. J. Måløy [2], M. Bourgoin [1] & S. Santucci [1,3 ✉]

A pile of grains, even when at rest in a silo, can display fascinating properties. One of the most celebrated is the Janssen effect, named after the pioneering engineer who explained the pressure saturation at the bottom of a container filled with corn. This surprising behavior arises because of frictional interactions between the grains through a disordered network of contacts, and the vessel lateral walls, which partially support the weight of the column, decreasing its apparent mass. Here, we demonstrate control over frictional interactions using ferromagnetic grains and an external magnetic field. We show that the anisotropic pairwise interactions between magnetized grains result in a radial force along the walls, whose amplitude and direction is fully determined by the applied magnetic field. Such magnetic Janssen effect allows for the fine tuning of the granular column apparent mass. Our findings pave the way towards the design of functional jammed materials in confined geometries, via a further control of both their static and dynamic properties.

[1] Univ Lyon, ENS de Lyon, Univ Claude Bernard, CNRS, Laboratoire de Physique, Lyon, France. [2] PoreLab, The Njord Centre, Department of Physics, University of Oslo, Oslo, Norway. [3] Lavrentyev Institute of Hydrodynamics, Siberian Branch of the Russian Academy of Sciences, Novosibirsk, Russia. ✉ email: stephane.santucci@ens-lyon.fr

Granular media are ubiquitous in everyday life, in both natural and man-made systems. Their handling is of prime importance for a wide range of major industrial sectors, from civil engineering, to mining, agriculture, food, chemical and pharmaceutical industries. Around 40 billion of tonnes of sand and gravels are extracted every year for the building industry, while 2.5 billion of tonnes of iron ores and more than 2 billion of tonnes of cereals are produced annually[1]. These colossal quantities which keep on increasing with dramatic environmental consequences explain why granular matter appears as the second most manipulated material after water[2]. Even the smallest improvement in our understanding of particulate materials may have profound economic and societal impacts. However, granular media still confront engineers and physicists with challenging fundamental questions, which over the last decades has triggered an upsurge of studies within a very active research community[2–6]. They can display unusual physical properties, common to a wide range of amorphous materials, such as foams, emulsions and gels. These Soft Jammed Materials exhibit a peculiar dual mechanical behavior arising from their complex disordered microstructure: they behave as solids at rest, while they unjam and flow like liquids above a critical yield stress[6–9].

Here, we focus on the simplest static configuration, commonly encountered in our daily lives: a container filled with a granular assembly up to a given height. For sufficiently tall column (higher than the vessel diameter), the pressure at its bottom saturates at a value much smaller than the hydrostatic pressure that one would measure for a Newtonian liquid column. In order to understand and avoid the failure of silos, which is still an industrial problem nowadays, Janssen proposed in a seminal work[10] a simple continuum phenomenology to explain this puzzling behavior. His model relies on three hypothesis, (i) within the grains packing, the vertical stresses $\sigma_z$ are redistributed proportionally to the horizontal ones, $\sigma_r = k\sigma_z$, with a phenomenological constant $k$ (ii) the frictional contact forces $F_z$ between the particles and the walls are at their Coulomb threshold, $F_z = \mu F_r$ with $\mu$ the friction coefficient between the grains and the wall, and (iii) the grain assembly is considered as a continuous medium. These assumptions lead to the description of a mass screening: the apparent mass $m_J = m_\infty[1 - \exp(-m_0/m_\infty)]$ saturates exponentially at a value $m_\infty = (\lambda/h)m_0$, with the characteristic length scale $\lambda = 2R/(4\mu k)$, $2R$ being the silo diameter, and $m_0$ the total mass of grains. Despite some criticisms and refinements[5,9,11–14], Janssen's approach has become a "classic", and its predictions have been shown to accurately describe experimental data, as long as the packing preparation leads to a full mobilization of the frictional forces along the container wall.

We propose here to revisit this classic problem by extending it to the case of ferromagnetic particles. One can then easily control and tune the interactions between the grains of the packing (using an external magnetic field), in contrast for instance to the influence of the particles shape and roughness on the frictional properties[15,16], or to the cohesive forces induced by capillary bridges in a humid environment[17–21], or by the occurrence of triboelectric charges[22,23], which are all much more difficult to control. Similar systems have already been used to modify the packing fraction and repose angle of a granular pile[24–28], as well as its flowing behavior in a silo or in a rotating drum[27,29]. When applying an external magnetic field **B**, ferromagnetic particles acquire a magnetization with a dipolar moment $\mathbf{d} = V\chi_m/\mu_0\mathbf{B}$, with $V$ the volume of the grain, $\mu_0$ the magnetic permeability of vacuum and $\chi_m$ the volume susceptibility of the grain. Therefore, each particle $i$ of the packing will interact with any other particle $j$ via the dipole-

dipole interactions[30]:

$$\mathbf{f_{ij}} = \frac{3\mu_0}{4\pi r_{ij}^5}\left(d^2\left(1 - 5\cos^2\alpha\right)\mathbf{r_{ij}} + 2dr_{ij}\mathbf{d}\cos\alpha\right), \quad (1)$$

with the interaction potential, $U_{ij} = \frac{\mu_0 d^2}{4\pi r_{ij}^3}(1 - 3\cos^2\alpha)$, where $\alpha$ is the angle between the applied magnetic field and the separation vector $\mathbf{r_{ij}}$ between grains $i$ and $j$. Interestingly, as illustrated on Fig. 1a, the induced magnetic dipolar pair interaction displays an anisotropic multi-polar field, either repulsive or attractive depending on the relative position of the grains, and the applied magnetic field direction. The dimensionless number that compares the strength of the magnetic interactions with the weight of the grains $\Psi = \frac{\chi_m^2 B_0^2}{\mu_0 a\rho g}$, with $\chi_m$ the magnetic susceptibility (ceiling at 3 for spherical grains assembly[31]), quantifies the involved interactions[28,32].

## Results

Two series of measurements were performed with a vertical and a horizontal magnetic field. Figure 2a, b show the apparent measured mass as a function of the actual total mass of poured grains $m_0$ for various amplitude $B_0$ of the applied field, quantified by the parameter $\Psi$, which in our experiments ranges from 1.5 to ~35. First, without any magnetic field applied, $\Psi = 0$, we retrieve the typical exponential saturation of the measured mass predicted by Janssen[10]. Fitting our experimental data in the absence of magnetic field by Janssen' expression of $m_J$, with the saturation mass $m_\infty$ as a single free parameter, we obtain $m_\infty = 48$ g, characteristic of our

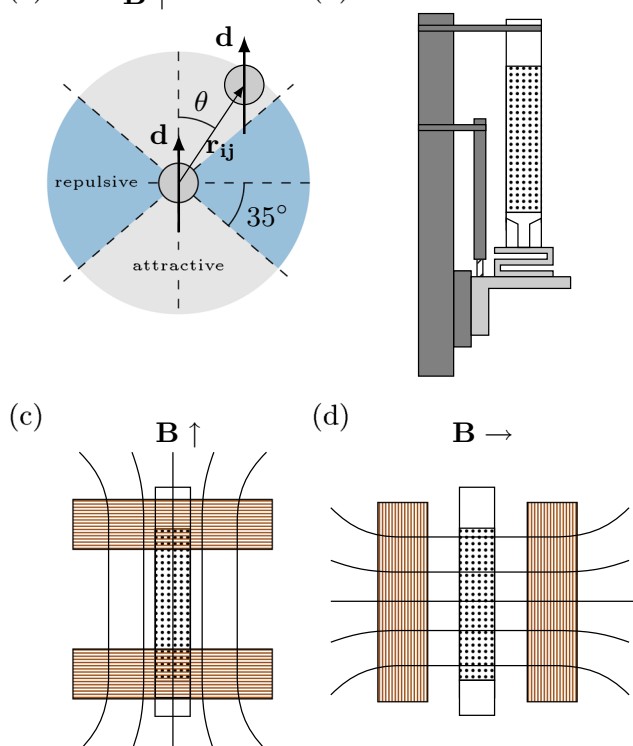

**Fig. 1 Experimental setup. a** Anisotropic magnetic dipolar interaction between two ferromagnetic particles subjected to the magnetic field **B**. **b** Ferromagnetic particles fill a cylindrical tube and rest on a piston fixed to a force sensor. This piston can be displaced vertically thanks to a motorized translation stage. **c**, **d** The column is placed between two magnetic coils mounted in a Helmoltz vertical or horizontal configuration to create a uniform magnetic field.

granular column. This fit is provided in Supplementary Information. We observe a deviation from Janssen's prediction as soon as a magnetic field is applied to the granular column. This is particularly evident for a vertical field and tall columns for which we observe a decrease of the packing apparent mass, as the amplitude of the magnetic field is increased. For small applied amplitudes the measured mass is still well described by Janssen's model, although with a decreasing saturating mass $m_\infty$ and thus a larger phenomenological constant $k$, quantifying the radial redistribution of vertical stresses within the packing. For high amplitudes of the magnetic field, a striking more is less effect is observed. For tall columns, adding more grains lowers the measured mass!

The impact of a horizontal magnetic field on the apparent mass of the packing is less pronounced, although we can still observe a small but systematic increase of the measured mass for tall columns with $\Psi$. This could be interpreted, within the Janssen approach, as a reduction of the radial redistribution of stresses.

These results may appear counter-intuitive. Indeed, one may expect particles to reorganize, tending to align and form chains in the direction of the applied field. This would potentially increase the force on the lateral walls for horizontal fields, while increasing the vertical force and the apparent mass for a vertical applied field. However, using a transparent glass tube we detected no such rearrangement nor motion of the particles .

In order to analyze more deeply the deviations between our measurements and the classical Janssen prediction, we show in Fig. 2c the mass difference $m - m_J$ for a series of experiments performed at the maximum amplitude of the magnetic field $\Psi = 35$. Interestingly, while we clearly observe again the strong

deviation to the classical exponential saturation for tall columns, with a lower (resp. larger) apparent mass for a vertical (resp. horizontal) magnetic field direction, we notice another striking behavior, where the effect reverses, for rather short granular columns, with a mass $m_0 \lesssim 30$ g, corresponding to a height typically smaller than the silo diameter. Indeed, when applying a vertical magnetic field to short columns, we observe an overshoot, where the measured mass appears slightly larger (peaking at 30–40% of $m_0$ for the highest values of magnetic field we explored) than the actual mass of the packing. This can be seen in Fig. 2d, where the apparent mass $m$ is renormalized by the actual mass of the packing $m_0$, with the magnetic field fixed at its maximum amplitude. Such a puzzling magnetic reverse Janssen effect is reminiscent of recently reported results[33], where a delicate sequential filling protocol of a narrow container leads to the emergence of compressional frictional forces with conventional (non magnetic) beads. The present study goes beyond this, by indicating the possibility to control the reverse Janssen phenomenon, without requiring any specific filling protocol[34,35], via tunable particle-particle magnetic interactions. With horizontal magnetic fields, for short columns, we instead obtain an undershoot, with an apparent mass smaller than the actual mass of the packing. Our results and more specifically our model (described below) suggest that this overshoot presents a maximum, which occurs for a column height of only a few particle diameters. The high density of the ferromagnetic particles renders a thorough study difficult. In contrast, the reverse Janssen effect reported in[33] with less dense particles, is found maximum for columns heights exceeding ten particle diameters. These observations suggest interesting strategies to potentially finely tune this reverse effect, by investigating the combined role of the applied field, the tube to particle diameter and the particles density.

**Model**. To rationalize our measurements, we first need to consider all the pairwise dipole-dipole interactions within the ferromagnetic granular packing. For a given particle $j$, $f_j^\beta$, with $\beta = r, \theta, z$, gives respectively the radial, azimuthal and vertical components of the magnetic force resulting from its dipolar interaction with all the other particles. The components of the total net force associated with the magnetic effects on the whole packing, of height $h$, are given by $F^\beta(h) = \sum_i f_i^\beta$, where the sum is taken over all the particles of the packing. Due to the cylindrical symmetry of the set-up, $f_j^\theta = 0$, so that there is no net azimuthal force. For the vertical component, the mirror symmetry of the system with respect to the mid-plane ($z = h/2$) means that contributions of two symmetric particles $j$ and $j'$ compensate each other ($f_j^z = -f_{j'}^z$) and have a zero net effect on the global force. Thus, magnetic interactions do not add any global net vertical force on the packing. The only global effect of magnetic interactions on the packing is on the radial component. Such radial effects can be anticipated to be maximal near the walls of the container. Indeed, for a particle $j$ placed along the symmetry axis of the container ($r_j = 0$), symmetry constraints impose that $f_j^r = 0$, as radial magnetic pair interactions of a particle on the axis with its symmetric neighbors average to zero. Conversely, if the particle $j$ is off-axis, a radial net force $f_j^r$ emerges due to the symmetry breaking related to the finite-size of the silo, as illustrated in Fig. 3a.

Those magnetic interactions can be computed numerically using Eq. (1) and the positions of particles within a packing generated by a Discrete Element Method simulation[36,37]. Figure 3b shows a typical spatial $(r,z)$ map of the radial component of the magnetic force (averaged over the angular coordinate $\theta$), for a 10 cm high column, under a vertical applied

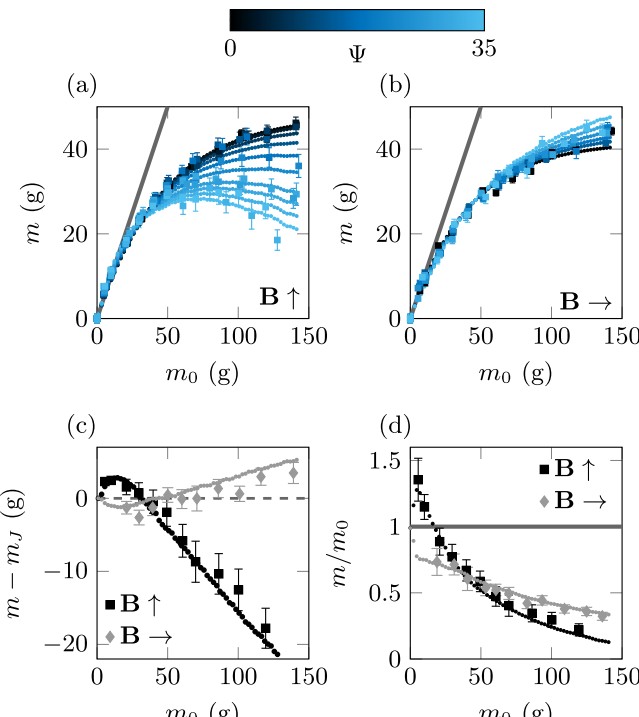

**Fig. 2 Apparent mass measurements and predictions. a**, **b** Mass measurement of the bead packing for different field intensity, characterized by $\Psi$, and direction. The gray line corresponds to the hydrostatic limit. For a field amplitude $\Psi = 35$, **c** gives the difference between the mass measurement and Janssen prediction $m_J$; while **d** displays the mass measurement renormalized by the true mass of the column $m_0$, the gray line corresponds to the hydrostatic limit, $m = m_0$. Our theoretical model (points reported on the various panels) reproduces quantitatively our experimental measurements (square symbols).

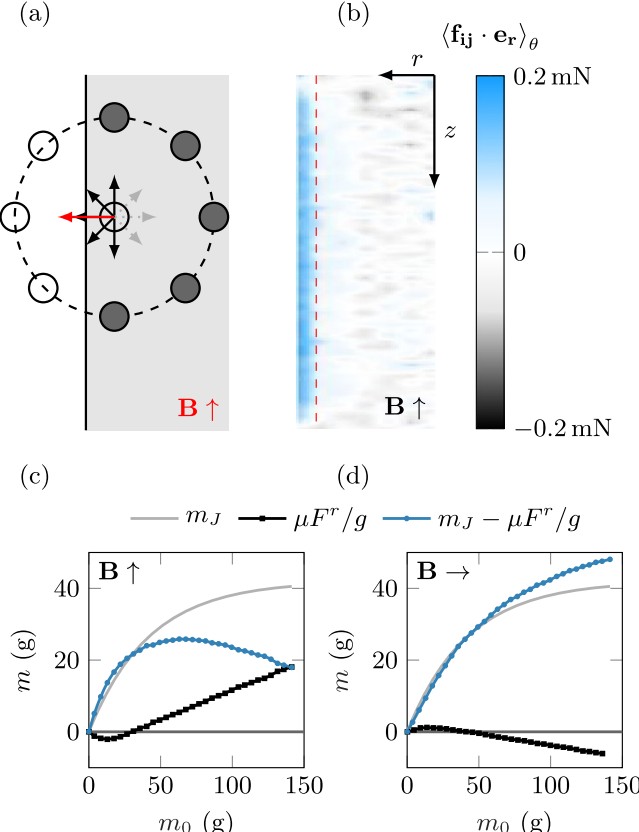

**Fig. 3 Magnetic Janssen effect model. a** The magnetic force on a particle within the packing, but off-the vertical axis, induced by neighboring ones (in dark gray), is non-zero and along the radial direction, due to the tube finite-size, and the non-compensation of the forces from the missing particles (in white). For a vertical magnetic field, the resulting radial force is repulsive (red arrow), as confirmed by the right panel. **b** Numerical computation of a spatial ($r,z$) map of the magnetic pair-interaction radial component (averaged over the coordinate $\theta$) for a 10 cm column, submitted to a magnetic field of $\Psi = 35$. Model curves for each direction of the magnetic field (**c** vertical, **d** horizontal). The light gray line shows Janssen prediction; the black squares line is the result of the computed magnetic mass loss $\mu F^r/g$; the blue circles line (difference of both curves) gives the predicted apparent mass of the column.

field ($\Psi = 35$). In this case, we observe that the magnetic pair interactions result in a radial repulsive force concentrated at the walls. Furthermore, this radial force appears constant along the vertical position $z$, independent of the column height, except at the top and bottom due to the edge effect (see Supplementary Material for more details). As a result, for sufficiently tall columns $h > 2R$, we expect a linear dependence of the global radial force $F^r$ with the packing mass $m_0$, as observed in Fig. 3c. For short columns $h < 2R$, corresponding to a packing mass $m_0$ smaller than 30 g, edges effects are dominant. Nevertheless, for this case of a vertical magnetic field, the magnetic interactions are found to be mostly attractive (see Fig. S6 in Supplementary Material), resulting in a negative global radial force $F^r$. Repulsive and attractive magnetic interactions compensate and cancel out for a column height around $h = 2R$.

Finally, we follow Janssen's approach in order to estimate the contribution of this radial force to the effective mass of the packing. We consider the granular packing as a cohesive continuous medium, and the particles wall contacts at the Coulomb limit, such that the radial net magnetic force $F^r(h)$ can be associated to a global vertical contribution to the pressure

measured at the bottom of the silo, $F(h) = -\mu F^r(h)$. The minus sign accounts for the fact that a positive (outwards) radial contribution corresponds to a stronger force supported by the walls, hence reducing the apparent mass of the column. Such "magnetic Janssen effect" can therefore be either compressive or tensile depending on the sign of $F^r$, leading to a tunable –enhanced or hindered– apparent mass of the ferromagnetic packing, $m = m_J - \frac{\mu}{g}F^r(h)$.

Computing the global radial force $F^r(h)$ for various packing heights, the resulting predicted apparent mass is displayed in Fig. 3c, d (blue curve), for a vertical and horizontal magnetic field (of a fixed amplitude, $\Psi = 35$ and a friction coefficient $\mu = 0.40 \pm 0.05$) as a function of the actual granular mass. The contribution from the classical Janssen effect and the additional magnetic effect are also shown. Figure 2 demonstrates that our theoretical approach reproduces quantitatively all the experimental measurements, without any adjustable parameter.

## Discussion

Importantly, our model predicts that for sufficiently tall columns (typically taller than the silo diameter), the magnetic contribution to the apparent mass evolves linearly with the actual added mass $m_o$. In particular, when a vertical field is applied, the apparent mass decreases linearly with $m_o$, pointing towards the existence of an invisibility threshold corresponding to a critical total mass $m_0^c(B_0)$ (dependent on the applied field amplitude) above which the apparent mass vanishes and the column becomes undetectable at the bottom of the silo. Alternatively, for a given total mass $m_0$, invisibility can be reached by adjusting the amplitude of the magnetic field above a critical amplitude $B_0^c(m_0)$.

Figure 4a shows how this invisibility threshold $m_0^c$, computed numerically, evolves with the amplitude of the magnetic field, given by the dimensionless number $\Psi$. We observe that this critical mass $m_0^c$ decreases as a power-law with $\Psi$, $m_0^c \propto \Psi^{-1}$, with an exponential cut-off at large magnetic field (corresponding also to small granular columns).

We can furthermore predict such scaling behavior, following our derived expression of the apparent mass of the ferromagnetic granular column, with a given applied field,

$$m = m_J - \frac{\mu}{g}F^r(m_0^c, \Psi) = 0 \qquad (2)$$

The total radial magnetic force $F^r$ is directly proportional to the magnetic field strength dimensionless number $\Psi$. Furthermore, as already mentioned, this magnetic force increases linearly with $m_0$, $F^r \propto (m_0 - m_0^{2R})$, where $m_0^{2R} \simeq 30$ g the actual mass of the granular column of height $h = 2R$, the silo diameter (see Fig. S6 in Supplementary Material). Considering tall columns ($h \gg 2R$), the Janssen mass has reached its saturation value, $m_J \simeq m_\infty$, and the actual mass of the packing $m_0$ is thus larger than $m_0^{2R}$, such that, $F^r(m_0^c, \Psi) \propto (m_0^c - m_0^{2R})\Psi \simeq m_0^c\Psi$. We therefore obtain the following scaling for this critical mass, $m_0^c \propto m_\infty \frac{g}{\mu}\Psi^{-1}$, in excellent agreement with our numerical computation.

The predicted values of this critical mass $m_0^c$ are beyond the experimentally reachable ones, set by the limits of our current set-up: the maximum intensity of the external magnetic field $B_{\max} \simeq 170$ G, (leading to $\Psi \simeq 35$) is fixed by the maximum electrical power that the coils can handle, while granular columns taller than $h_{\max} \simeq 10$ cm, (which corresponds to a mass around 150 g) experience inhomogeneous applied field. These experimental limitations are displayed on the Fig. 4a, b. Nevertheless, Fig. 4 a shows that the invisibility condition ($\Psi \simeq 35, m_0^c \simeq 250$ g) is not so far from the experimental limits of our apparatus. We have therefore performed a single apparent mass measurement for a very tall column ($h_{\max} \simeq 15$ cm) corresponding to an actual

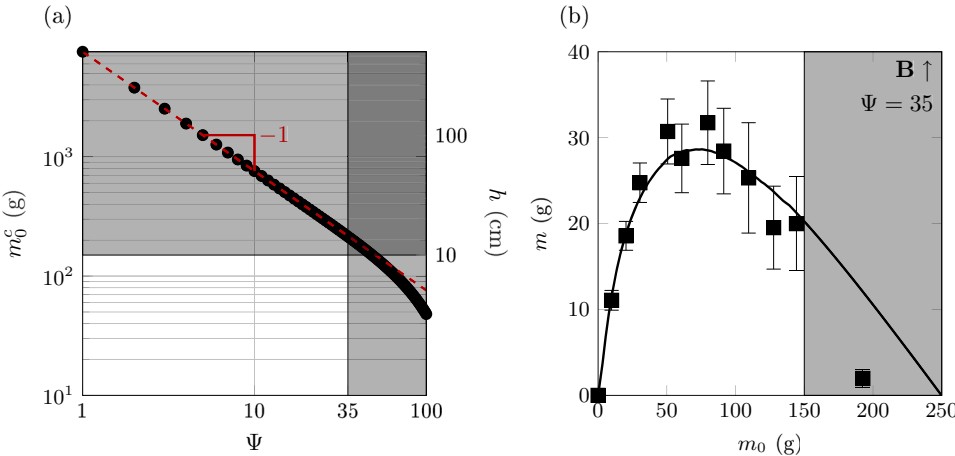

**Fig. 4 Invisibility threshold and experimental limitations. a** Numerical computation of the critical mass $m_0^c$, for which the apparent mass of the packing is zero, as a function of the magnetic field strength, given by the dimensionless number $\Psi$. This critical mass decreases as a power-law, $m_0^c \propto \Psi^{-1}$, as predicted theoretically. The gray zones represent the mass and magnetic field intensity that cannot be reached by our experimental apparatus. **b** Apparent mass measurement for the highest magnetic field intensity, applied vertically, and comparison to our model (black line).

mass around 200 g, with a vertical magnetic field of maximum amplitude ($\Psi \simeq 35$), presented in Fig. 4b. Strikingly, we observed that the granular column becomes invisible with an apparent mass close to zero in these conditions. This is in qualitative agreement with our model. Indeed, this critical mass appears slightly smaller than our theoretical prediction, $m_0^c = 250$ g, probably due to the fact that the applied magnetic field for such a tall granular column is not perfectly homogeneous all along the packing.

To conclude, the present study reveals a magnetic Janssen effect leading to versatile control of the apparent mass of a ferromagnetic granular column. Beyond such remarkable static properties, being able to finely tune the apparent mass of a granular column opens an appealing perspective towards novel dynamical properties, with for instance the control of the discharge of silos. Finally, we note that our study can be extended to the case of composite granular media (where only a fraction of the grains are ferromagnetic) with an even broader range of possible applications.

## Methods

The setup shown in Fig. 1 consists of a copper tube of 19.57 mm internal diameter filled with steel beads of radius $a = 0.75$ mm and density $\rho = 8000$ kg m$^{-3}$. The beads lay on a Polytetrafluoroethylene (PTFE) piston of diameter 18.30 mm carefully aligned with the tube to ensure no contact with its walls. The piston is attached to a force sensor which can measure the apparent mass of the packing with a precision of $\pm 0.10$ g. We followed the protocol described in[11,38] to fully mobilize the frictional forces along the tube: the piston is mounted on a vertical translation stage displacing the beads assembly downwards over a distance $\Delta h = 7$ mm at a velocity $v = 0.2$ mm s$^{-1}$, see Supplementary Materials for more details. The packing apparent mass is then obtained by measuring its mean value during the displacement, while the standard deviation around this value reflects the typical measurement dispersion. We placed the whole setup between two magnetic coils in a Helmholtz configuration, creating a uniform magnetic field, either vertical or horizontal depending on the respective orientation of the coils. The magnetic field is turned on and applied to the granular column, right after the packing preparation, just before its mass measurement. With this setup, we can reach a maximum external magnetic field of $B_{max} \approx 170$ G, leading to the dimensionless number $\Psi \approx 35$. The inner diameter of our magnetic coils (20 cm) sets the maximum height of the granular column, for which the applied field can be considered uniform, to $h_{max} = 10$ cm, which corresponds to a mass around 150 g. For a uniform magnetic field, the individual induced dipolar moments **d** can be considered identical for all beads and aligned with the direction of the field.

## Data availability

The data that support the findings of this study are available from the corresponding author upon reasonable request.

## Code availability

The code used for discrete particle simulations is the open-source package MercuryDPM[36].

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

## Acknowledgements
We thank L. Vanel, J-C. Géminard, and N. Taberlet for enlightening discussions. We thank the Research Council of Norway through its Centre of Excellence funding scheme, project number 262644, the support of ENS de Lyon, and of the CNRS, through the French-Norwegian IRP, (D-FFRACT). S.S. acknowledges also the support of the Russian Government with grant no. 14.W03.31.0002.

## Author contributions
All authors significantly contributed to this work as a team effort. S.S. and M.B. designed the study. L.T. performed the experiments and the theoretical analysis, supervised by S.S., M.B. and K.J.M.. S.S. and L.T. wrote the first draft of the manuscript. All authors read critically and participate to the writing of the manuscript.

## Competing interests
The authors declare no competing interests.
