## [Peer Review File · Nature Communications]

Reviewers' Comments:

Reviewer #1:

Remarks to the Author:

A long time ago, Janssen rationalized that the apparent mass at the bottom of a column of grains is smaller than the mass of the added grains, due to the contribution of frictional forces between the grains and the container wall. The manuscript "Magnetic Janssen Effect" reports the experimental investigation of this effect, in the presence of magnetic particles. In the set-up considered by the authors, therefore, a new control parameter influences the experimental results: the applied magnetic field. The manuscript investigates and rationalizes the effect of the magnetic field on the apparent mass.

The manuscript is of fundamental interest, and tackle and issue of great practical relevance. The results are sound and well presented, and I, therefore, recommend publication in nature communications.

There is, however, an inconsistency, or lack of clarity, that needs to be resolved. Fig. 2c reports a plot of $m - m_J$, with m_J the apparent mass as estimated by the Janssen model. It is not clear how this m_J is estimated. Indeed, according to Janssen's model, the apparent mass should saturate, while it is clear from Fig.2a that this is not always the case in the experimental set-up considered by the authors.

Related to this point, the authors claim that they observe a "reverse Janssen effect". I am not convinced that this is the case, as Fig. 2a suggests that the apparent mass is always smaller than the added one, as clear from the comparison with the hydrodynamic limit. I guess the authors interpret their result as a reverse effect as they focus on the extrapolated m_J . This point must be clarified.

I also notice that the manuscript stresses in the introduction that the reported magnetic effect allows for the tunability of the apparent mass. This tunability indeed occurs within the set-up considered by the authors. However, the considered set-up is not typical in industrial settings. In particular, this set-up involves the preparation of the packing at constant applied magnetic field B . I would therefore strongly suggest discussing, and possibly experimentally investigating, how the apparent mass prepared according to some protocol depends on B . This investigation might prove "tunability" in a more direct way.

Massimo Pica Ciamarra

Reviewer #2:

Remarks to the Author:

This paper describes a beautiful piece of research. While most of the introduction focuses on granular media as bulk commodities (measured in tonnes of material), I think that the potential impact of the findings lies actually somewhere else, namely when granular media are filled into confined geometries. For those situations, I fully agree with the authors that their results may "pave the way towards the design of functional jammed materials, via a further control of both their static and dynamic properties".

The experiments are elegant and appear to be done expertly. The paper is written very well and the results are described clearly. I have just a few comments for the authors to consider.

1. In Figure 2c it seems to me that the model predictions slightly overestimate the measured values for larger m_0 . Based on the data shown, the slopes of the model predictions are slightly steeper in positive direction for horizontal applied field and in negative direction for vertical field than what the measured data suggest. It would be extremely informative to show measured data

and model results for fields larger than 150 grams. The authors state that their apparatus can go to at least 190 grams (to check for the invisibility threshold), so I would very much like to see data plotted out to that mass value for m_0 . In fact, a plot of $(m_0)_c$ versus $(B_0)_c$ would be very informative.

2. Figure 3 provides the central results and especially panel 3c hints at the apparent mass (blue trace) tending toward zero (= invisibility threshold) for even larger m_0 than shown. The authors state that this very remarkable phenomenon, namely the absence of detectable mass at the bottom of a column, happens at $m_0 = 190\text{g}$ under the conditions in their experiments. It therefore would be important to show data beyond 150g and all the way out beyond 200g.

3. A few minor items:

a) The authors write that "Janssen's approach has become a "classic", taught nowadays to every bachelor student in Physics". I have to, unfortunately, disagree with the second part of that sentence: Janssen's approach is not taught to physics students at any university I am familiar with (because the physics of granular media does not appear in the curriculum), and furthermore is not even taught to most mechanical or chemical engineering students.

b) on page 3 the authors write "One can then easily control and tune the interactions between the grains of the packing (using an external magnetic field), in contrast for instance to the influence of the particles shape and roughness on the frictional properties [18, 19], or to the cohesive forces induced by capillary bridges in a humid environment [20–24], or by the occurrence of triboelectric charges [25, 26], which are all hardly tunable." The term "hardly tunable" actually conveys the meaning of "not tunable", as in "this is hardly correct" = "this is incorrect". I do not think the authors have this in mind, since their point is the contrast with tunability via the magnetic field. I would suggest replacing 'hardly tunable' with 'much more difficult to control or adjust'.

c) Bottom of page 5: "Formild applied amplitudes" should be "For small applied amplitudes"

d) page 6 bottom: 'a striking "the more is less" effect is observed' should be 'a striking "more is less" effect is observed'

Dear Editor,

We are pleased to resubmit a revised version of our manuscript.

We would like to thank the two Referees, first for their very positive comments and remarks about our work, but more importantly, for their relevant questions and suggestions, which helped us to improve the presentation of our results.

Please find below our detailed point-by-point answer to each referee's remarks and questions, that we have carefully considered, together with a version of our revised manuscript, where both modifications and new materials are highlighted in blue. Notice also that we have slightly modified our manuscript to satisfy the specific format of Nature Communications (reducing the abstract length, introducing some sections and subsections, notably). We hope that you will find this new version of the manuscript suitable for publication in Nature Communications.

Best regards,

Stéphane Santucci, on behalf of the authors

Reviewer #1 (Remarks to the Author):

A long time ago, Janssen rationalized that the apparent mass at the bottom of a column of grains is smaller than the mass of the added grains, due to the contribution of frictional forces between the grains and the container wall. The manuscript "Magnetic Janssen Effect" reports the experimental investigation of this effect, in the presence of magnetic particles. In the set-up considered by the authors, therefore, a new control parameter influences the experimental results: the applied magnetic field. The manuscript investigates and rationalizes the effect of the magnetic field on the apparent mass.

The manuscript is of fundamental interest and tackle an issue of great practical relevance. The results are sound and well presented, and I, therefore, recommend publication in nature communications.

First, we would like to thank the referee, Massimo Pica Ciamarra, for his constructive remarks and very positive comments about our work.

There is, however, an inconsistency, or lack of clarity, that needs to be resolved. Fig. 2c reports a plot of $m - m_J$, with m_J the apparent mass as estimated by the Janssen model. It is not clear how this m_J is estimated. Indeed, according to Janssen's model, the apparent mass should saturate, while it is clear from Fig. 2a that this is not always the case in the experimental set-up considered by the authors.

We thank the referee for pointing out this possible source of misunderstanding, which actually is not an inconsistency in our results, but a lack of clarity in our presentation. In the new version of the manuscript, we have modified and introduced a few sentences (detailed below) in order to clarify this aspect of our analysis and to avoid any misunderstanding.

As explained in the manuscript, m_J corresponds to Janssen's prediction of the apparent mass of the granular column, $m_J = m_\infty \left(1 - e^{-\frac{m_0}{m_\infty}} \right)$, which saturates exponentially at a value m_∞ .

Without any magnetic field applied, $\Psi = 0$, our mass measurements are indeed very well described by the classical Janssen approach (see the black symbols on Figure 2 a & b, and also the new figure provided in Supplemental Material). Fitting those experimental data by this expression, with the saturation mass m_∞ as the single free parameter, we obtain the value of $m_\infty \simeq 48g$, for our granular column. Following Janssen expression, the corresponding value of m_J is then computed as a function of m_0 : it characterizes the apparent mass of our granular column as a function of the added mass, in the absence of a magnetic field, and is a fixed curve throughout our study. Notice also, related to the

last comments of the referee (see below), that we have checked the robustness of our measurement of m_J , by changing our protocol (for instance, turning off the magnetic field during the experiment).

Then, as soon as a magnetic field is applied to the granular column, a systematic deviation to the exponential saturation predicted by Janssen is clearly observed in Figures 2 a & b, as the amplitude of the magnetic field increases. Actually, the curves reported in Figure 2c focus on such deviations, giving simply $m - m_J$ the difference between our mass measurement of the granular column submitted to a magnetic field (of maximum amplitude, $\Psi = 35$), and its apparent mass without any magnetic field applied (which again is perfectly well predicted by Janssen model).

Furthermore, it is also important to remark that in our model, this mass difference is directly related to the total magnetic radial force F^r : $m - m_J = -\frac{\mu}{g} F^r$

We added the following sentences in the main text of the manuscript and the following figure together with its description, in Supplemental Information, to clarify our analysis: "Fitting our experimental data in the absence of magnetic field by Janssen's expression of m_J , with the saturation mass m_∞ as a single free parameter, we obtain $m_\infty = 48$ g, characteristic of our granular column. This fit is provided in Supplemental Information."

Related to this point, the authors claim that they observe a "reverse Janssen effect". I am not convinced that this is the case, as Fig. 2a suggests that the apparent mass is always smaller than the added one, as clear from the comparison with the hydrodynamic limit. I guess the authors interpret their result as a reverse effect as they focus on the extrapolated m_J . This point must be clarified.

We thank the referee for this remark, which helped us clarifying this aspect in our manuscript. As a matter of fact, we do indeed observe in our experimental data a "magnetic reverse Janssen effect", for the case of vertically applied fields and small granular columns. In order to show more clearly such effect, and furthermore demonstrate that it is independent of our measurement of m_J , we have modified our presentation, showing now in Figure 2d the apparent mass of the granular column simply renormalized by its actual mass, m/m_0 , as a function of m_0 (as done in [34]), for the highest magnetic field amplitude we could reach with our set-up $\Psi = 35$. In the former Figure 2d, we were showing the difference between the mass measurement and Janssen prediction, renormalized by the difference with the true mass of the column $(m - m_J)/(m_0 - m_J)$, whose interpretation was not as direct.

Our new figure is easier to read: for small granular columns (below 20 g), we clearly observe that for a vertical magnetic field, the apparent mass is larger than the actual mass of the grains packing, $m > m_0$.

This overshoot concerns only small granular columns, and even though it can reach about 30%, for the highest magnetic field amplitude, its absolute value of a few grams is therefore difficult to directly observe on Fig. 2a (where the full vertical scale is about 50 g). A zoom on the data of Fig. 2a for small values of m_0 clearly shows however the excess of mass compared to the hydrostatic when a large vertical field is applied.

This reverse effect is also well predicted by our model (see for instance figure 3c). Interestingly, our model also suggests that for very small values of m_0 the hydrostatic limit may be recovered. The “magnetic reverse Janssen effect” we observe therefore shares qualitative similarities with the “reverse Janssen effect” recently discovered and reported in [34], while adding a certain tunability thanks to the magnetic field, without requiring a particular filling protocol.

However, physical constraints of our experimental configuration prevent currently a further detailed characterization. Indeed, this effect (and possibly a maximum of this overshoot) is observed for small masses m_0 , reached within a few layers of beads. Experimentally, exploring in detail the low mass regime with a significant number of layers would require low density ferromagnetic grains. Numerically, one could imagine simulating artificial situations of light grains with a strong magnetization. Alternatively, the case of mixtures of dense ferromagnetic particles with light non-ferromagnetic particles could be considered. All these are paths which will be explored in future dedicated studies. Indeed, these observations open interesting perspectives of further dedicated studies (beyond the scope of the present work) to fully characterize the “magnetic reverse Janssen effect” and its possible similarities and differences with the “classical” one [34]. In particular, a systematic study of the role of the applied field, of the tube to particle diameter and of particles density shall guide new discoveries and strategies to finely tune this reverse effect.

In the new version of the manuscript, we therefore clarify the existence of a “reverse magnetic Janssen effect”, emphasizing on the qualitative similarities and differences with the recently reported results of the “reverse Janssen effect” in [34].

“Indeed, when applying a vertical magnetic field to those short columns, we can observe an overshoot, where the measured mass appears slightly larger than the actual mass of the packing, as emphasized by Figure 2d, where the apparent mass m is renormalized by the actual mass of the packing m_0 , for the maximum amplitude of the magnetic field.

Such a puzzling “magnetic reverse Janssen effect is reminiscent of recently reported results [34], ...

...

Our results and more specifically our model (described below) suggest that this overshoot presents a maximum, which occurs for a column height of only a few particle diameters. The high density of the ferromagnetic particles renders a thorough study difficult. In contrast, the “reverse Janssen effect” reported in [34] with less dense particles, is found maximum for columns heights exceeding ten particle diameters. These observations open interesting strategies to potentially finely tune this reverse effect, by investigating in detail the combined role of the applied field, the tube to particle diameter and the particles density.

I also notice that the manuscript stresses in the introduction that the reported magnetic effect allows for the tunability of the apparent mass. This tunability indeed occurs within the set-up considered by the authors. However, the considered set-up is not typical in industrial settings. In particular, this set-up involves the preparation of the packing at constant applied magnetic field B . I would therefore strongly suggest discussing, and possibly experimentally investigating, how the apparent mass prepared according to some protocol depends on B . This investigation might prove “tunability” in a more direct way.

The ability to tune the apparent mass of the granular column is indeed an important (and probably the main) result of our work which can be of practical relevance, as noticed by both referees.

First of all, we would like to clarify an aspect of our measuring protocol that the Referee misunderstood. Our set-up does not involve the preparation of the packing at a constant applied field. On the contrary, as described in Supplementary Information, the particles are poured inside the silo, without any magnetic field applied. The magnetic field is turned on right before moving the piston downward for the mass measurement.

In order to avoid any misunderstanding, we now also mention in the main text of the new version of the manuscript this important aspect of our protocol:

“The magnetic field is turned on and applied to the granular column, right after the packing preparation, just before its mass measurement.”

Our measuring protocol follows the one proposed in [14, 36] to obtain robust mass measurement and notably, ensuring that the frictional forces along the container wall are fully mobilized (respecting Janssen’s hypothesis). Of course, we could have modified the packing preparation, as done in [34, 38], but our present work goes beyond, by demonstrating the possibility to control the apparent mass of the granular column without requiring any specific filling protocol, but thanks to controllable magnetic interactions between the particles.

In our experiments, the magnetic field is kept constant during the mass measurement. As the reviewer seems to suggest, and as we put forward in the last sentence of the abstract of our manuscript, varying the magnetic field during our mass measurement would be very interesting and could lead to novel results. However, it goes clearly beyond the scope of our current study, and as such will be the subject of future dedicated work. Nevertheless, as a brief first test, we could check that during one mass measurement when turning-off the magnetic field, the apparent mass suddenly jumps back to the value predicted by the classical Janssen effect. A typical example is shown on the figure below, for a granular assembly of mass $m_o = 40$ g, with in our silo an apparent mass of $m_J = 28.2$ g without any magnetic field applied, and, when applying a magnetic field of maximum amplitude ($\Psi = 35$), an apparent mass of $m = 20.7 \pm 2.3$ g. As soon as, the magnetic field is switched-off during the mass measurement, the apparent mass jumps back to the value $m = 28.4 \pm 0.9$ g, equal to m_J .

This observation on one hand confirms the robustness of our measurements and, furthermore, demonstrates our ability to tune directly the apparent mass of the ferromagnetic granular column.

We added a subsection *“1.4 Direct tunability of the apparent mass of the column”* in the Supplementary Material to explain and report such observation.

We thank again the referee for his constructive remarks which helped us to improve the clarity of our presentation and overall quality of our manuscript.

Reviewer #2 (Remarks to the Author):

This paper describes a beautiful piece of research. While most of the introduction focuses on granular media as bulk commodities (measured in tonnes of material), I think that the potential impact of the findings lies actually somewhere else, namely when granular media are filled into confined geometries. For those situations, I fully agree with the authors that their results may “pave the way towards the design of functional jammed materials, via a further control of both their static and dynamic properties”.

The experiments are elegant and appear to be done expertly. The paper is written very well, and the results are described clearly. I have just a few comments for the authors to consider.

First, we would like to thank the referee for the very positive comments about our work and relevant remarks and suggestions; we are very grateful for such appreciation.

The first remark about the impact of our findings, concerning “the confined geometries” is indeed particularly relevant; we actually made it clear in the new version of the manuscript (specifically in the abstract). Let us now answer the following comments, below:

1. In Figure 2c it seems to me that the model predictions slightly overestimate the measured values for larger m_0 . Based on the data shown, the slopes of the model predictions are slightly steeper in positive direction for horizontal applied field and in negative direction for vertical field than what the measured data suggest.

We can indeed observe in Figure 2c some slight deviations between some of our experimental measurements and the predictions of our model, in particular for the largest values of m_0 (and hence for the taller columns) we have explored. We point however that this slight discrepancy concerns few points and remains within experimental error-bars. Several reasons can explain those small deviations:

- For tall columns, the granular packing may start experiencing the non-homogeneity of the applied magnetic field, due to the finite size of the Helmholtz coils (discussed in detail below, in the next point addressed by the referee, regarding the invisibility threshold). Such inhomogeneity effects are not taken into account in our model. Because of this experimental limitation, it is expected that some discrepancy (which remains small though in our results) may appear for the tallest columns.
- On the other hand, one can also notice that the slope in the large m_0 limit of the curves shown in Figure 2c is directly related to the friction coefficient μ between the particles and the walls of the silo. Throughout our study, we have kept its value fixed and equal to 0.40, in agreement with reported values from the literature. However, this friction coefficient may have a slightly different value depending on specificities of the grains' material, which would modify accordingly those slopes: $m - m_j = -\frac{\mu}{g} F^r$.

Keeping μ as free parameter, the figure on the left shows that if we fine tune the value of the friction coefficient, an almost perfect agreement between the model and this specific experimental data set can be found for $\mu = 0.35$.

However, we checked that the value $\mu = 0.40$ fits better our overall measurements, performed in various conditions (different column heights, different amplitude and direction of the magnetic field). In the new version of the manuscript, we nevertheless specify the error-bar on the friction coefficient: $\mu = 0.40 \pm 0.05$.

It would be extremely informative to show measured data and model results for fields larger than 150 grams. The authors state that their apparatus can go to at least 190 grams (to check for the invisibility threshold), so I would very much like to see data plotted out to that mass value for m_0 . In fact, a plot of $(m_0)_c$ versus $(B_0)_c$ would be very informative.

2. Figure 3 provides the central results and especially panel 3c hints at the apparent mass (blue trace) tending toward zero (= invisibility threshold) for even larger m_0 than shown. The authors state that this very remarkable phenomenon, namely the absence of detectable mass at the bottom of a column, happens at $m_0 = 190\text{g}$ under the conditions in their experiments. It therefore would be important to show data beyond 150g and all the way out beyond 200g.

We fully agree that “it would be very informative to show measured data and model results” for larger magnetic fields, as well as taller columns, with a mass larger than 150 grams, to check and study more thoroughly the invisibility threshold, that we just briefly discuss in the concluding section of the manuscript. Indeed, our results point towards the absence of a detectable mass at the bottom of the column, which is indeed a very remarkable phenomenon.

As suggested by the referee, we have now extended the discussion on this remarkable effect. In the new version of the manuscript, we provide now a theoretical prediction for this invisibility threshold $m_0^c(\Psi) \propto \Psi^{-1}$, that we could verify numerically:

Figure. 4a shows how this invisibility threshold m_0^c , computed numerically, evolves with the amplitude of the magnetic field, given by the dimensionless number Ψ . We observe that this critical mass m_0^c decreases as a power-law with Ψ , $m_0^c \propto \Psi^{-1}$, with an exponential cut-off at large magnetic field (corresponding also to small granular columns). We can furthermore predict such scaling behavior, following our derived expression of the apparent mass of the ferromagnetic granular column, submitted to a given applied field, $m = m_J - \mu/g Fr(m_0^c, \Psi) = 0$

The total radial magnetic force F^r is directly proportional to the magnetic field strength dimensionless number Ψ . Furthermore, as already mentioned, this magnetic force increases linearly with m_0 , $Fr \propto (m_0 - m_0^c)^2$, where $m_0^c \lesssim 30\text{g}$ the actual mass of the granular column of height $h = 2R$, the silo diameter (see fig. S3 in Supplemental Material). Considering tall columns ($h \gg 2R$), the Janssen mass has reached its saturation value, $m_J \simeq m_\infty$, and the actual mass of the packing m_0 is thus larger than m_0^c , such that, $Fr(m_0^c, \Psi) \propto (m_0^c - m_0^c)^2 \Psi \simeq m_0^c \Psi$. We therefore obtain the following scaling for this critical mass, $m_0^c \propto m_\infty/g \Psi^{-1}$, in excellent agreement with our numerical computation.

Regarding experimental measurements, as we explained in the Supplementary Material, we are facing an experimental difficulty here, preventing unfortunately such a detailed study. Actually, we realized that the part of the sentence “Pushing the limits of our experimental set-up” in the previous version of the conclusion of our manuscript may have been confusing and led to a misunderstanding.

We have now clarified this aspect, both in the discussion paragraph and in Supplementary Material, with dedicated paragraphs:

The predicted values of this critical mass are beyond the experimentally reachable ones, set by the limits of our current set-up: the maximum intensity of the external magnetic field $B_{\max} \simeq 170$ G, (leading to $\Psi \sim 35$) is fixed by the maximum electrical power that the coils can handle, while granular columns taller than $h_{\max} \sim 10$ cm, (which corresponds to a mass around 150 g) experience inhomogeneous applied field. These experimental limitations are displayed on the figures 4 a & b. Nevertheless, Figure 4a shows that the invisibility condition ($\Psi \sim 35$, $m_0^c \sim 250$ g) is not so far from acceptable experimental values. We have therefore performed a single apparent mass measurement for a very tall column ($h_{\max} \sim 15$ cm) corresponding to an actual mass around 200 g, submitted to a vertical magnetic field of maximum amplitude ($\Psi \sim 35$), reported on Figure 4b. Strikingly, we could indeed observe that, in these conditions, the granular column becomes invisible, with an apparent mass close to zero, in qualitative agreement with our model. Indeed, this critical mass appears slightly smaller than our theoretical prediction, $m_0^c = 250$ g, probably due to the fact that the applied magnetic field for such a tall granular column is not perfectly homogeneous all along the packing.

Indeed, with our current power supply and the size of our Helmholtz coils (which are already rather large, inner diameter of 20 cm), the maximum intensity of the external magnetic field we can reach is $B_{\max} \sim 170$ G, (leading to $\Psi = 35$) and the maximum height of the granular columns over which the applied magnetic field can be considered uniform is $h_{\max} \sim 10$ cm, which corresponds to a mass around $m_0 \sim 150$ g. This is the reason why we did not report detailed and systematic measurements above this value, fixed unfortunately by the limits of our current set-up. Indeed, for higher masses (taller columns), the magnetic field applied on the packing is not homogeneous anymore, introducing notably gradients, that will affect our experimental measurements.

Nevertheless, we also provide a unique apparent mass measurement close to zero, obtained for a very tall column ($h \sim 15$ cm) corresponding to an actual mass m_0 around 200 g, in qualitative agreement with our model. Indeed, in these conditions, the applied magnetic field is not homogeneous all along the column, which may explain why our measurement deviates from the theoretically predicted value, $m_0^c = 250$ g, for $\Psi = 35$.

3. A few minor items:

We thank the Referee for her/his careful reading of the manuscript; We have followed all the various modifications suggested:

a) The authors write that “Janssen’s approach has become a “classic”, taught nowadays to every bachelor student in Physics”. I have to, unfortunately, disagree with the second part of that sentence: Janssen’s approach is not taught to physics students at any university I am familiar with (because the physics of granular media does not appear in the curriculum), and furthermore is not even taught to most mechanical or chemical engineering students.

In the new version of the manuscript, we have now removed this part of the sentence, “taught nowadays to every bachelor student in Physics”. As a matter of fact, this is something we do teach to our students at ENS de Lyon in some optional lectures, although this is very likely due to self-research-interest of the professors in the department :)

b) on page 3 the authors write “One can then easily control and tune the interactions between the grains of the packing (using an external magnetic field), in contrast for instance to the influence of the particles shape and roughness on the frictional properties [18, 19], or to the cohesive forces induced by capillary bridges in a humid environment [20–24], or by the occurrence of triboelectric charges [25, 26], which are all hardly tunable.” The term “hardly tunable” actually conveys the meaning of “not tunable”, as in “this is hardly correct” = “this is incorrect”. I do not think the authors have this in mind, since their point is the contrast with tunability via the magnetic field. I would suggest replacing ‘hardly tunable’ with ‘much more difficult to control or adjust’.

Yes, we fully agree, and we thank the referee for pointing out this language mistake; the text has been modified accordingly.

c) Bottom of page 5: "Formild applied amplitudes" should be "For small applied amplitudes"
Yes, we have now corrected this typo.

d) page 6 bottom: 'a striking "the more is less" effect is observed' should be 'a striking "more is less" effect is observed'
Yes, the text has been modified accordingly.

Again, we thank the referee for her/his relevant and useful remarks, which helped us improving and enriching our manuscript.

Reviewers' Comments:

Reviewer #1:

Remarks to the Author:

The authors have clearly addressed my concerns and have revised the manuscript to improve clarity. I am happy to support the publication of the article in its present form.

Reviewer #2:

Remarks to the Author:

I read the authors' response to the comments from the first round of reviews and find that they have very satisfactorily answered all questions raised. Indeed, they performed additional measurements and added to the discussion of the invisibility threshold, thereby improving an already excellent the paper. I am recommending publication in its present form.